# Open access resource for cellular-resolution analyses of corticocortical connectivity in the marmoset monkey

Piotr Majka [1,2,3], Shi Bai [2,3], Sophia Bakola[2,3,4], Sylwia Bednarek [1], Jonathan M. Chan[2,3], Natalia Jermakow[1], Lauretta Passarelli [4], David H. Reser[3,5], Panagiota Theodoni [6,7,8], Katrina H. Worthy[3], Xiao-Jing Wang [6,10], Daniel K. Wójcik [1,10], Partha P. Mitra [9,10] & Marcello G.P. Rosa [2,3,10 ✉]

Understanding the principles of neuronal connectivity requires tools for efficient quantification and visualization of large datasets. The primate cortex is particularly challenging due to its complex mosaic of areas, which in many cases lack clear boundaries. Here, we introduce a resource that allows exploration of results of 143 retrograde tracer injections in the marmoset neocortex. Data obtained in different animals are registered to a common stereotaxic space using an algorithm guided by expert delineation of histological borders, allowing accurate assignment of connections to areas despite interindividual variability. The resource incorporates tools for analyses relative to cytoarchitectural areas, including statistical properties such as the fraction of labeled neurons and the percentage of supragranular neurons. It also provides purely spatial (parcellation-free) data, based on the stereotaxic coordinates of 2 million labeled neurons. This resource helps bridge the gap between high-density cellular connectivity studies in rodents and imaging-based analyses of human brains.

[1] Laboratory of Neuroinformatics, Nencki Institute of Experimental Biology of the Polish Academy of Sciences, 3 Pasteur Street, 02-093 Warsaw, Poland. [2] Australian Research Council, Centre of Excellence for Integrative Brain Function, Monash University Node, Clayton, VIC 3800, Australia. [3] Biomedicine Discovery Institute and Department of Physiology, Monash University, Clayton, VIC 3800, Australia. [4] Department of Biomedical and Neuromotor Sciences Pharmacy and Biotechnology, University of Bologna, Bologna 40126, Italy. [5] Graduate Entry Medicine Program, Monash Rural Health-Churchill, School of Rural Health, Monash University, Churchill, VIC 3842, Australia. [6] Center for Neural Science, New York University, NY 10003, USA. [7] New York University - East China Normal University Institute of Brain and Cognitive Science, Shanghai 200241, China. [8] New York University Shanghai, Shanghai 200122, China. [9] Cold Spring Harbor Laboratory, Cold Spring Harbor, New York, NY 11724, USA. [10]These authors jointly supervised this work: Xiao-Jing Wang, Daniel K. Wójcik, Partha P. Mitra, Marcello G.P. Rosa. ✉email: marcello.rosa@monash.edu

We introduce a resource comprising curated results of retrograde tracer injections performed in the cortex of marmosets (*Callithrix jacchus*), made accessible through a web site (http://marmosetbrain.org). This resource builds upon earlier demonstrations of the feasibility of registration of retrograde tracer data in primates to a common template, and of sharing such data through online platforms[1,2], but represents a distinct step towards cellular-level connectomic analyses of the primate brain. The present release incorporates methodological refinements which improve the accuracy of the attribution of neurons to areas, as well as new datasets including the stereotaxic coordinates of neurons revealed by the experiments, estimates of the white matter distances involved in each connection, and volumetric representations that enable integration with neuroimaging platforms. This functionality is made available through a dedicated analysis interface which enables quantification and visualization of results in individual cases, or aggregated by cortical area.

The contributions of neurons to perception, action, or cognition are fundamentally determined by the connections they form with other neurons. Complete maps of neuronal connections have been achieved for simple nervous systems[3], but uncovering the organizational principles that govern neuronal connectivity in mammals is most realistically tackled at the mesoscopic level[4], by taking advantage of regularities in the patterns of connections formed by clusters of adjacent neurons[5]. Although this is still a vast challenge, progress in informatics has dramatically increased our capabilities to generate, store, and analyze large datasets, which in turn enhance our understanding of brain function through detailed computational models and analyses.

Presently, there are several projects aimed at exploring connections throughout the mouse brain using grids of anterograde and retrograde tracer injections, which yield single cell resolution and information about the direction of information flow[6,7]. In parallel, the use of magnetic resonance imaging has allowed studies of the principal axonal tracts in the human brain, but at a much lower resolution[8]. Integrating these two streams of investigation has been difficult, given not only the different technologies used, but also neuroanatomical differences, particularly with respect to the cerebral cortex. Several subdivisions of the primate frontal and posterior parietal cortex, which are involved in higher-order cognition and sensorimotor integration, do not exist as separate areas in rodents[9–11]. Likewise, the primate occipital and temporal lobes comprise a large number of visual and auditory association areas which have no clear rodent counterparts[12,13]. Thus, cellular-resolution datasets of connections in non-human primates are necessary to bridge the gap between the gold standard tracer-based datasets obtained in rodent brains and our evolving knowledge of the human brain.

The most extensive resources on cortical connections in non-human primates currently available are CoCoMac[14] (an online platform based on meta-analysis of the literature) and CoreNets (a collection of tracer injections in 29 areas, obtained with consistent methodology)[15,16]. Both of these give access to information on the macaque monkey cortex, but are limited in terms of access to full datasets and integrated functionality for analysis. A recently available dataset offers high-resolution images of tracing experiments in marmosets, but no facility for quantification[2]. Here we introduce a resource which significantly extends present capabilities, by giving open access to complete, curated datasets obtained in marmosets, integrated with visualization and quantification tools. Marmosets are small (~350 g) New World primates which have become increasingly important for studies of cognitive function and dysfunction, in part due to the relatively short developmental cycle, which facilitates the development of transgenic lines[17] and studies across the life span[18].

The cerebral cortex is conceptualized to consist of areas characterized by distinct cytoarchitecture and myeloarchitecture, as well as different distributions of immunocytochemical markers and patterns of gene expression. Neurons in different areas are traditionally regarded as distinct in terms of connections and functions. In principle, the clustering of neurons into areas simplifies the problem of detecting regularities in cortico-cortical patterns of connections, and thus the information flow; indeed, each of the neuroinformatics resources currently available for connections in the mammalian brain is anchored on the notion of areas. However, the reality is far more complex. Functionally defined areas can be both histologically and connectionally heterogeneous[19–21], and single cytoarchitectural areas often prove, upon closer examination, to include connectionally distinct subregions[22,23]. Further, the histological limits of cortical areas are often very subtle[24]: for example, current estimates of the number of areas in the macaque cortex vary between 91[15] and >150[25,26]. These factors hinder integration of data raised by different groups, and constitute obstacles for scientific progress in general: the different sets of nomenclatures and criteria, together with the fact that the primary datasets are not made publicly available, typically make it impossible to analyze results retrospectively.

These challenges define the theoretical, analytical, and computational requirements for a resource aimed at facilitating research on the connectional architecture of the cortex. Primarily, such a resource should provide access to data in the form of directional connections within a common stereotaxic space, using a consistent nomenclature to describe the results both spatially and semantically (thus enabling analyses based purely on spatial distribution of connected cells, as well as those according to areas). The underlying data should be provided in full, and the resource should offer support for large-scale models and simulations by enabling access to data in a programmatic and a machine-mineable way[27,28]. The Marmoset Brain Connectivity Atlas (http://marmosetbrain.org) addresses these goals. To enable graph-based network analyses, this resource includes tools for the quantification of connections between currently recognized areas[29]. In addition, it comprises the stereotaxic coordinates of labeled neurons, thus opening new avenues of exploration that are purely spatially based, enabling direct comparisons with results of non-invasive imaging methods, and allowing for future analyses that incorporate new schemes of parcellation. We showcase the capabilities of this resource by performing convergent area, injection, and single-cell based tests of hypotheses derived from human neuroimaging experiments, which postulate relationships between neuronal connectivity, distance from primary sensory and motor areas, and affiliation to different resting state networks[30,31].

## Results

**Distribution of tracer injections**. The Marmoset Brain Connectivity Atlas is accessible through the http://marmosetbrain.org portal. This resource provides online access to the primary experimental data, as well as the database of connectivity patterns quantified according to the parcellation of the marmoset cortex[29] (see Supplementary Table 1 for a list of areas and their abbreviations, and Supplementary Table 2 for a list of tracers used). The present release includes the results of injections centered in 55 of the 116 currently recognized areas of the marmoset cortex, encompassing subdivisions of prefrontal, premotor, superior temporal, parietal, and occipital complexes (Fig. 1). A list of all injections is available in Supplementary Table 3, while comparisons between data obtained with different tracers are provided in Supplementary Figs. 1 and 2. Across all injections, 1,968,388

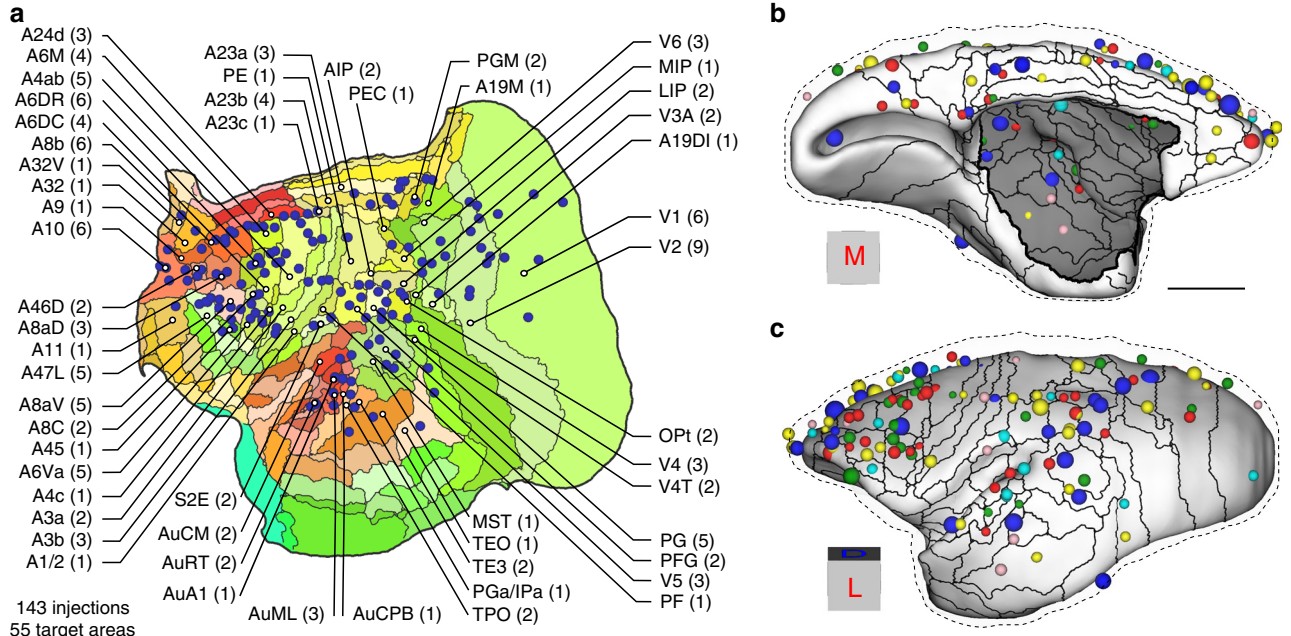

**Fig. 1 Locations of the 143 tracer injections registered to a template based on the reference atlas. a** The injections are illustrated in a two-dimensional (flat) map of the marmoset cortex, with the number of injections in each area indicated. For abbreviations of the areas, see Supplementary Table 1. Medial **b** and lateral **c** mid-thickness projections on the template brain, with the locations of the injections indicated (the dashed outline indicates the pial surface). In these views, the tracers used in each injection are coded by color (blue: FB, yellow: DY, dark green: FE, red: FR, teal: CTBgr, pink: CTBr), and the diameters of the spheres are proportional to the volumes of the injection sites. Scale bar: 5 mm.

tracer-labeled cells were mapped. The data were registered to a volumetric brain template through a pipeline that assures anatomical accuracy by taking into account cytoarchitectural, myeloarchitectural, and chemoarchitectural information to guide the registration process (see the "Methods" section, 3D reconstruction and mapping procedure). This, and expert curation of the results, provides a high degree of spatial precision, largely circumventing the effects of individual variability.

**The marmoset brain connectivity portal**. Figures 2–4 illustrate the main features of the portal. The connectional data are accessible via the Injections page (Fig. 2a), where the locations of tracer injections are displayed in a computationally generated unfolded map of the marmoset cortical areas (Fig. 1a). A searchable index allows the user to select injections based on single or combined criteria, including area, type of tracer used, case designation, or metadata keywords. Access to the data obtained following an individual injection can also be achieved by pointing and holding the mouse over a location on the map; this takes the user to a summary of the data, and gives access to navigating the full dataset for that injection.

Selecting an injection opens a section viewer, which displays the results of all injections placed in a given cerebral hemisphere (Fig. 2b). The viewer allows visualization of the locations of the neurons labeled by different tracers, the core and halo regions of each injection, and high-resolution images of the underlying histology, which can be viewed at different magnifications. The identity of each cortical area, according to registration to the atlas parcellation[29], can be determined by hovering the cursor over the images, and results from individual tracers can be shown or hidden as required by using the switches on the top left of the navigator window. Rapid access to sections representing any anteroposterior level can be achieved by a slide-and-click navigator at the bottom of the page. Each injection is accompanied by metadata about the sex, age, and experimental history of each animal (accessible through the Case metadata

button), and summaries of the projection patterns can be obtained in the form of two-dimensional maps of the cortex (Flat Map button).

An important component of the Marmoset Brain Connectivity Atlas is the interface for exploring the quantitative results of the tracer injections, which can be accessed via the Connectivity Matrix tab. This part of the portal provides access to data on the strength and direction of connection between areas (fraction of labeled neurons, extrinsic; FLNe: Fig. 3a) and the laminar origin of the projections (fraction of supragranular layer neurons; SLN: Fig. 3b). In addition, estimates of distances between cortical areas (Supplementary Fig. 3) can be obtained through the Quick links menu (Fig. 4a). Each column in the matrices (Fig. 3) represents the connections of an area which received at least one tracer injection, which can be arranged according to alphabetical order, rostrocaudal coordinates, or hierarchical clustering (whereby areas characterized by similar connectivity patterns appear adjacent to each other). Alternatively, the connectivity data can be viewed as a graph (Graph view) which highlights spatial relations (Fig. 4a). Hovering the cursor over the intersection of two areas in the matrices, or pointing at a graph edge, provides information about the cases in which a specific connection was observed (Fig. 4b), and clicking reveals further information, including an average connectivity profile, interactive visualizations of the data for each injection (Fig. 4c) and metadata (Fig. 4d). Summaries of the data and detailed information (such as 3-d maps of distributions of labeled neurons compatible with neuroimaging platforms, and the stereotaxic coordinates of each labeled neuron) can be downloaded for each injection separately, collated into a master spreadsheet (Quick links menu, Fig. 4a), or accessed programmatically with a dedicated Application Programming Interface (http://analytics.marmosetbrain.org/wiki/api).

**Convergent area, injection, and cell-based analyses**. The Marmoset Brain Connectivity Atlas allows the analysis of global

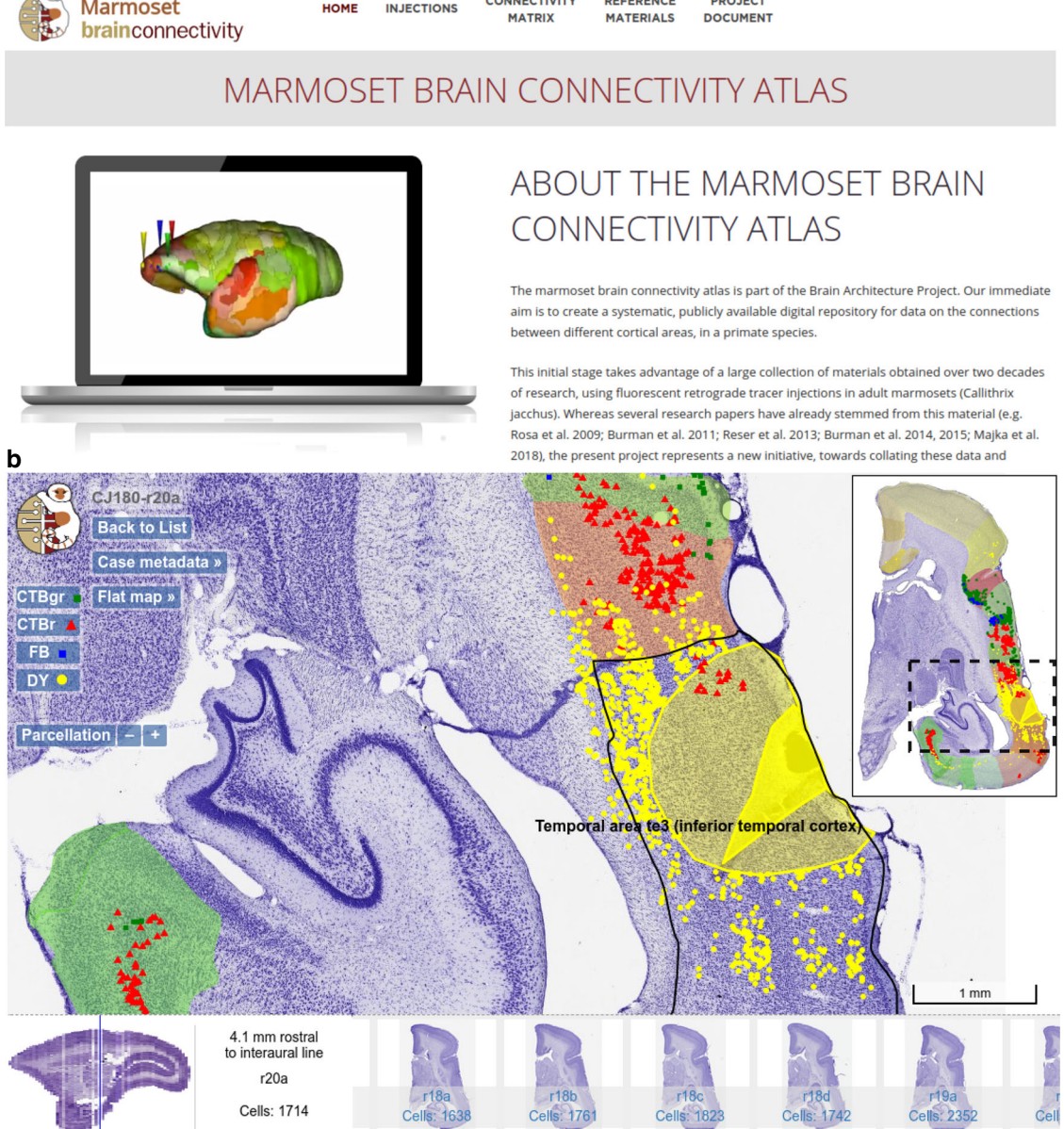

**Fig. 2 Overview of the Marmoset Brain Connectivity Atlas and high-resolution section viewer. a** Gateway page of the portal (http://marmosetbrain.org) offering access to different components of the website, including the primary experimental data (Injections tab) and quantitative results, such as the FLNe, SLN, and interareal distance matrices (Connectivity Matrix tab, see Fig. 3). The Reference Materials tab gives access to the reference atlas[29], the volumetric template of the marmoset brain, and histology protocols. The Project Document tab provides background information on the aims of the project and its implementation. **b** Highlight of the main features of the high-resolution section viewer (injection CJ180-DY used as an example). An image of a Nissl-stained section (r20) is overlaid with the injection site and halo (yellow polygons, drawn under the microscope), as well as locations of cells labeled by DY and other tracers injected in this case (points in various colors). The navigation bar (bottom) offers a quick way of traversing across the dataset while the widgets (buttons on the top left) allow for adjusting the view according to one's requirements. The cortical areas on the section are annotated based on the registration to the reference atlas[29] (thumbnail in the top right corner). The contents of the portal are available under an open license (https://creativecommons.org/licenses/by-sa/4.0/).

properties of area to area connectivity, as well as finer-grained analyses based on the locations of neurons. This raises the opportunity to test the often-made assumption that network properties which emerge from analyses of interareal connectivity matrices[32–34] reflect properties of the underlying cellular networks. As an example, we tested the hypothesis that neurons in primary sensory and motor areas tend to receive shorter-ranged connections, in comparison with those located progressively farther from their boundaries. This was initially suggested based

on human functional connectivity, and later supported by analyses of macaque data[30,31,35], but not yet tested at the level of neuronal populations.

The first step was to test if marmoset data reflect the features observed in the macaque[31] using an analysis based on areas. In this analysis, the distance to the nearest primary area represents the distance from the injection site to the border of the nearest primary area (for injections placed within primary areas this distance is 0). The connectivity distance of an injection is

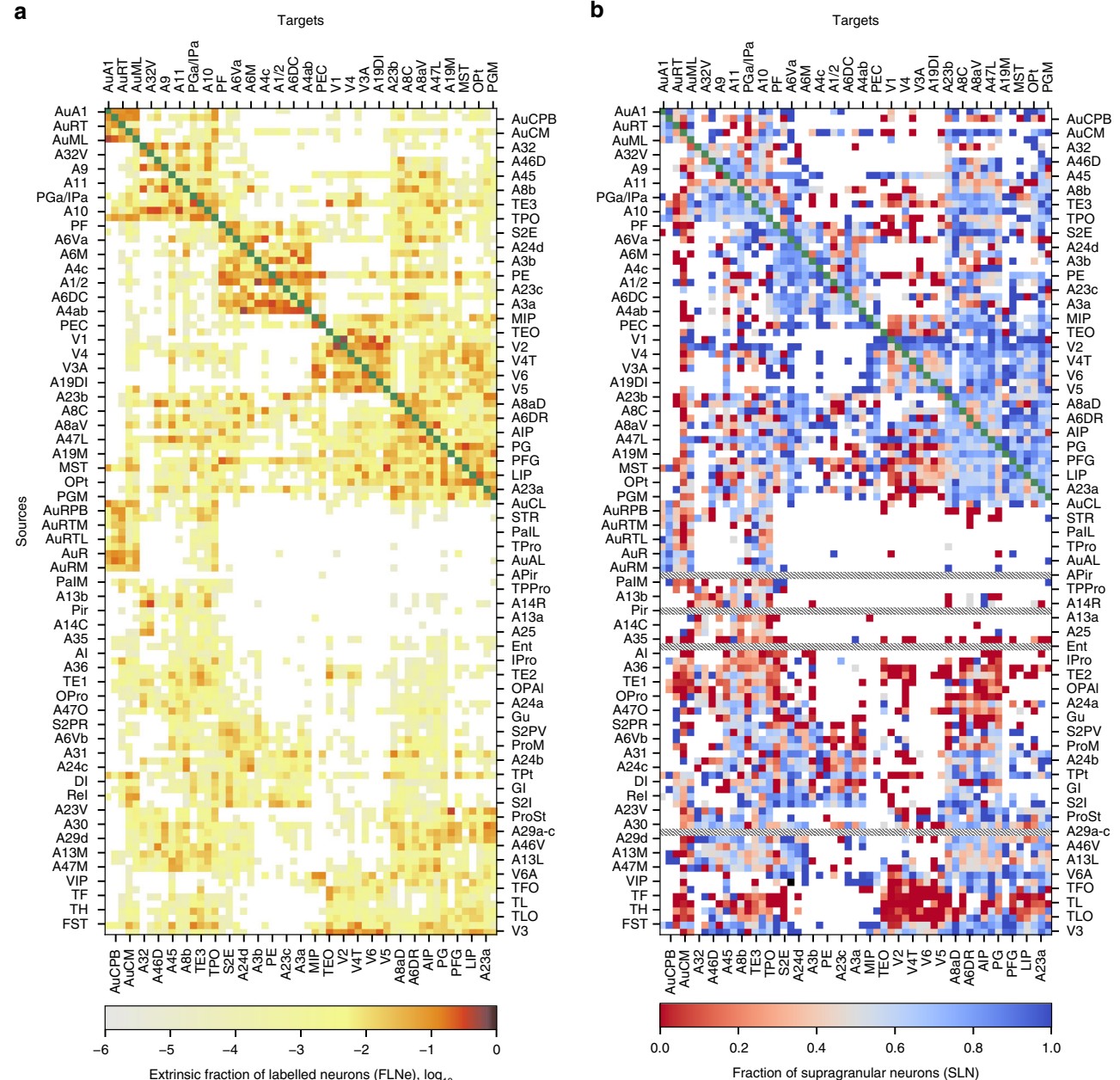

**Fig. 3 Cortico-cortical connection matrices available through the resource. The matrices are shown here with areas ordered according to hierarchical clustering. a** Connection strengths between areas (calculated as the mean FLNe). **b** The percentage of labeled neurons located in the supragranular layers (SLN). In these diagrams, each column represents the connections of an area that received at least one tracer injection, and the rows represent the projections from an area. The abbreviations of areas are provided in Supplementary Table 1. Blank cells indicate connections that were not detected, and green cells denote intrinsic connections (not evaluated in the matrices). Rows annotated with hatched lines on panel **b** correspond to areas in which cells were not divided into supra- or infragranular due to the lack of a visible layer 4 (i.e. the entorhinal cortex [Ent], piriform cortex [Pir], amygdalopiriform transition area [APir], and subdivisions of area 29 [A29a–c]).

equivalent to the sum of the distances between the centroids of the target area and source areas, multiplied by the number of labeled neurons found in the source area, and divided by the total number of labeled neurons found in all source areas. A general linear model (GLM) was used to assess the relation between the distance to the nearest primary area (independent variable) and connectivity distance (dependent variable). The results in marmoset (Fig. 5a) support the view that there is a significant increase in the average connection length received by an area as a function of its distance from the primary cortex (GLM,

$F_{3,143} = 12.92$, $p < 10^{-3}$, $\beta = 0.33$, 95% CI [0.21, 0.45]). Averaging the results for all injections into the same area[31] yielded similar results (Fig. 5b). The above analyses were based on the set of primary areas used in studies of human and macaque cortex[31,35]: V1, area 3 (A3a/A3b in the present nomenclature), the auditory core (AuA1, AuR, AuRT) and F1 (A4ab, A4c). As shown in Supplementary Fig. 4, the relationship between average connection length and distance to primary areas is robust irrespective whether the analysis is based on functionally defined primary sensory areas (e.g. including the gustatory cortex, Gu, and

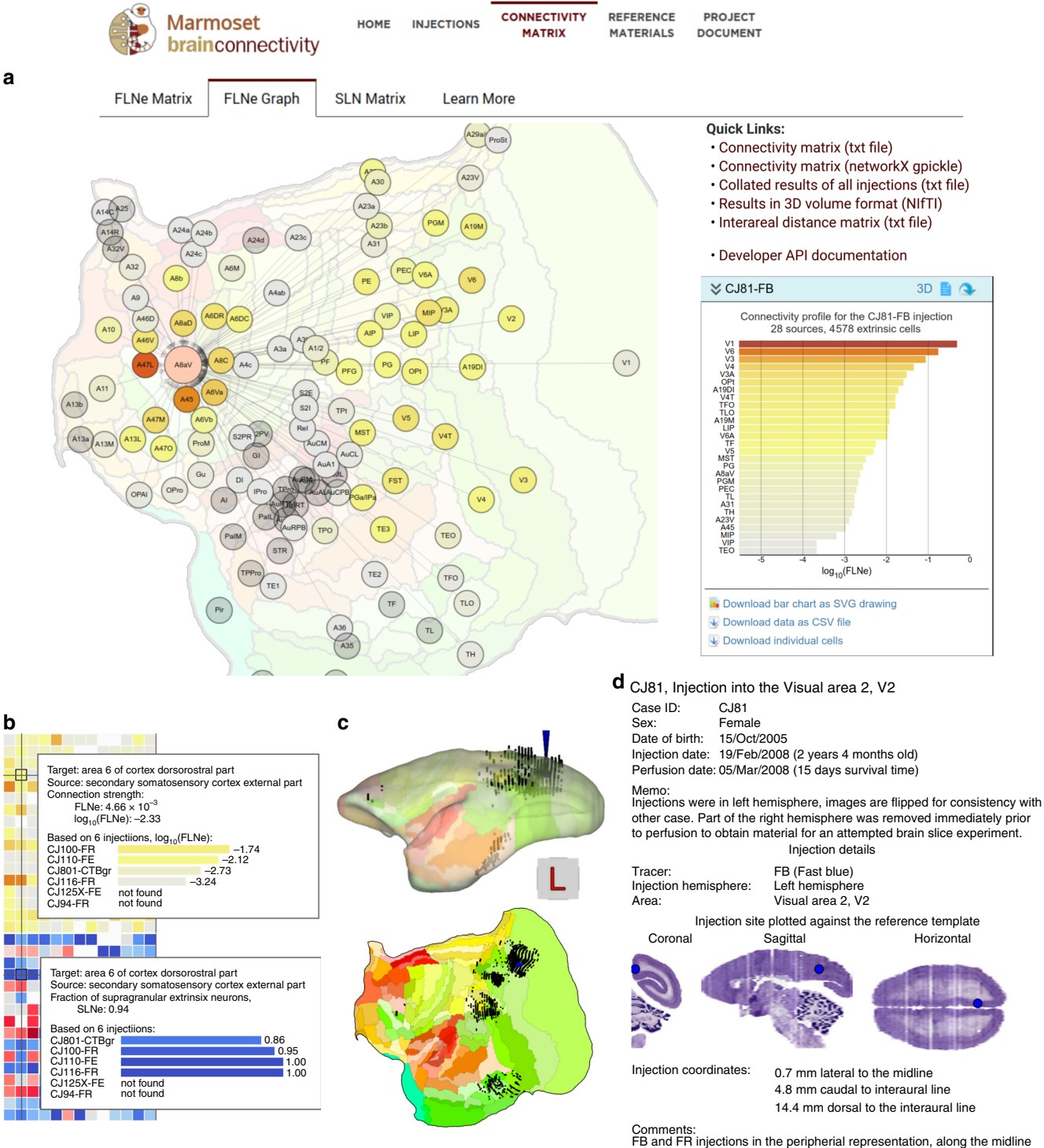

**Fig. 4 Interface for exploration of the quantitative results of retrograde tracer injections.** This interface is available at http://analysis.marmosetbrain. org. **a** Overview of the interface opened on the FLNe Graph tab, shown using an example area (area 8aV, part of the frontal eye field). **b** Closeups of the FLNe and SLN matrices (see Fig. 3 for the full view of both matrices) showing the results for the projection from the external segment of somatosensory area S2 (S2E) to the dorsorostral subdivision of premotor area 6 (A6DR; injections into A6DR). This view details the results of all injections into A6DR, including those in which projection from S2E to A6DR was not observed. **c** Visualizations of pattern of projections for an example injection (CJ81-FB, in visual area 2, V2) in a form of an interactive three-dimensional widget (top, lateral view) or two-dimensional (flat) map of the marmoset cortex (bottom). Black dots represent individual neurons labeled by the injection. **d** Metadata panel providing information about the animal as well as the details of each injection (here, metadata for injection CJ81-FB are presented). The contents of the website are provided under an open license (https://creativecommons. org/licenses/by-sa/4.0/).

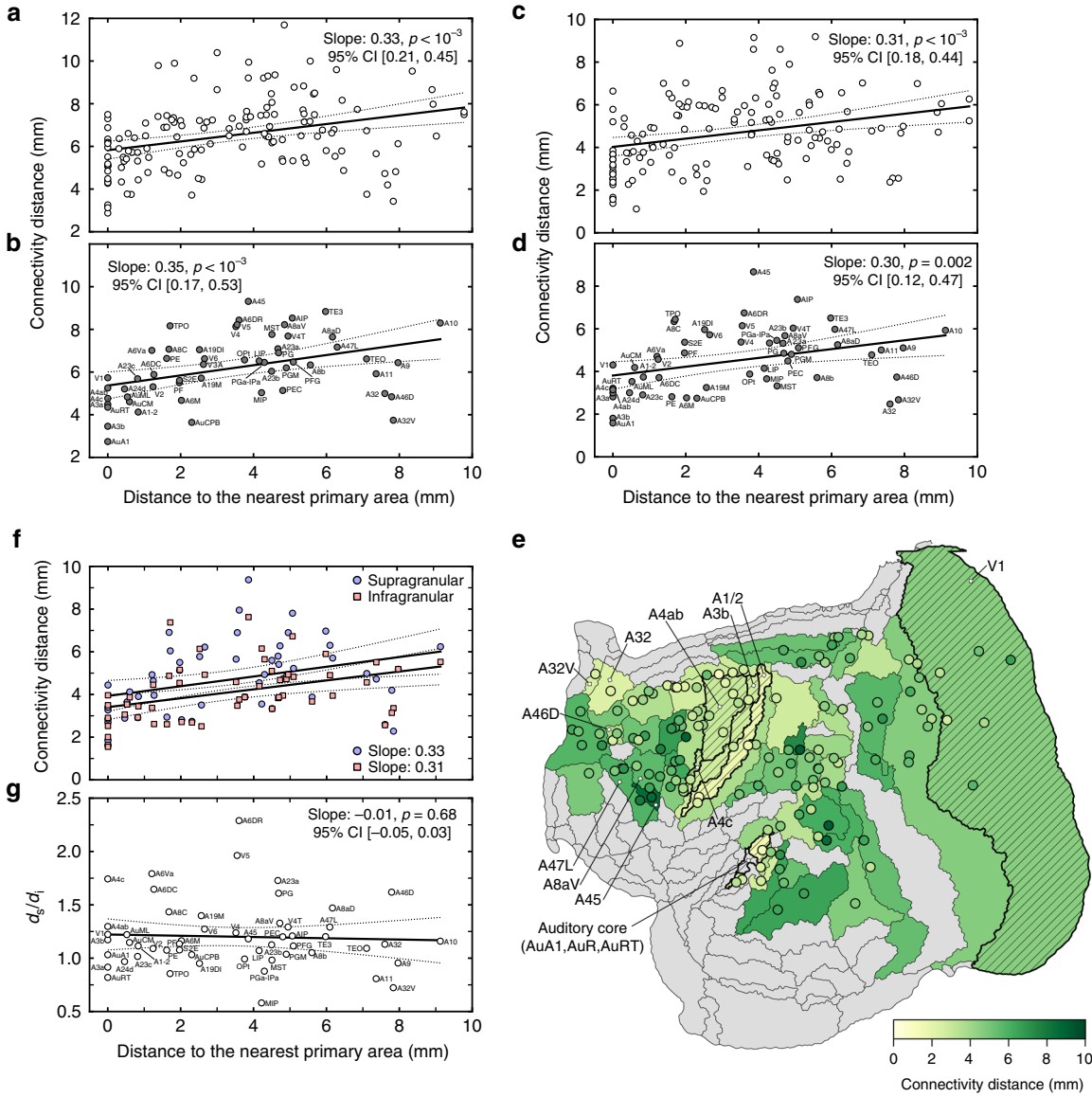

**Fig. 5 Average afferent connection length scales with distance from the borders of primary areas. a, b** Analyses in which the locations of individual labeled cells are approximated by locations of centroids of the corresponding areas, using a general linear model (GLM) that incorporated covariates related to locations of areas (areas near the center of the cortical sheet being expected to show on average shorter connections than those located near the edges) and their different volumes (larger areas being expected to show longer average connections when intrinsic connections are excluded). Analyses based on individual injections (**a**) or averages of the results of injections in the same area (**b**) reveal similar trends. **c, d** Analyses using the locations of individual neurons (both intrinsic and extrinsic). Here, the GLM also incorporated covariates to account for injection volume and the mean distance to every voxel annotated as a cortical area. **e** Visualization of the results of the cell-based connectivity distance on a two-dimensional map of the marmoset brain. The green tones of each circle indicate results of the injection-based analysis, and background colors the averaged connectivity distance for injections in an area. Diagonal hatch indicates the primary visual area (V1), areas A3a and A3b, the auditory core (AuA1, AuR, AuRT), and primary motor cortex (A4ab, A4c), and gray shading indicates areas where no injections are available. **f, g** Comparison of average the length of supragranular (blue circles) and infragranular (red squares) afferents to the same areas. The ratio of the average length of supragranular ($d_s$) and infragranular ($d_i$) afferents does not vary systematically with distance from primary sensory and motor areas. The abbreviations of areas are provided in Supplementary Table 1. Source data are provided as a Source Data file.

excluding the motor cortex and rostrotemporal auditory area, AuRT), or on koniocortex-type lamination (V1, AuA1, AuR, and A3b).

We next tested if a similar relationship was observable if the exact stereotaxic locations of the neurons are used, instead of approximating locations of labeled neurons with centroids of cortical areas. To avoid bias due to the distances of cells in different layers of the cortex to the nearest white matter, the locations of individual cells (and injections sites) were projected

to the nearest point on the mid-thickness surface of the template prior to other calculations. Here, the connectivity distance was estimated for each injection as an average length of simulated axonal tracts originating at the coordinates of each cell labeled by a given injection and terminating at the center of mass of the injection site. The pool of cells for each injection included both extrinsic and intrinsic labeled cells, but excluded those within a circular zone around the center of each injection. This zone, defined for each injection separately, was dictated by the volume

of the injection site and of the region immediately around it, in which individual neurons cannot be plotted reliably (the average radius of the exclusion zones in our sample was 515 μm; range 100–1100 μm). The results (Fig. 5c–e) demonstrate that the relationship between connectivity distance and distance to the nearest primary area, first detected by an area-based approximation, reflects a genuine property of the cellular circuits of the cortex (GLM, $F_{3,139} = 9.10$, $p < 10^{-3}$, $\beta = 0.31$, 95% CI [0.18, 0.44]), and that this relationship is present (Fig. 5f, g) both in connections formed by supragranular neurons (putative feedforward connections) and infragranular neurons (putative feedback connections)[15].

Further examination of the data presented in Fig. 5 reveals that prefrontal areas are quite heterogeneous in terms of average connection lengths, with rostral and medial prefrontal areas (areas 32, 32V, and 46D on Fig. 5e) showing increased emphasis on short-ranged connections relative to caudal prefrontal and ventrolateral prefrontal areas (e.g. areas 45, 47L, and 8aV). This is the case despite the data showing that in the marmoset, as in the macaque[36–39], all prefrontal areas receive long-range afferent connections, which originate in areas of superior temporal, retrosplenial, posterior parietal, and caudal orbital cortex (Fig. 6). The difference in connectivity distance (A32: 2.9 mm, A32V: 3.1 mm, A46D: 4.2 mm, against A45: 8.9 mm, A47L: 6.3 mm and A8aV: 5.9 mm) is likely to reflect the closer link of caudal and ventrolateral prefrontal areas to cognitive functions associated with the analysis of ongoing sensory input, which is conveyed by long-range projections from the occipital, superior temporal and posterior parietal regions[37,40,41]. In comparison, the main functions of rostral and medial prefrontal areas are more closely associated with internally generated states including memory, value, and goals.

The analysis illustrated in Fig. 5 also indicates that attributing single values to a cytoarchitectural area is unlikely to capture the local diversity of cortico-cortical connectivity. For example (Fig. 5e), injection sites in area 47L tended to show a larger emphasis on shorter connections rostrally (http://marmosetbrain.org, cases CJ71-FR [4.9 mm], CJ181-DY [3.8 mm]) than caudally (CJ73-FE, 7.9 mm). The capabilities introduced by the present resource provide a basis for future refinements of cortical parcellation schemes in the marmoset, when used in conjunction with other anatomical and physiological criteria.

**Local versus distant connectivity in cortical networks.** Large-scale networks characterized by co-activation measured by fluctuations in the BOLD signal have become an important concept in our current understanding of structure–function relationships in the mammalian cortex[42]. These networks, which can be mapped in the resting brain, are selectively activated during different tasks[43–45], and previous studies in non-human primates have established a correlation between these and groups of areas which are preferentially interconnected at the cellular, monosynaptic levels[46–48]. It has been hypothesized that the large-scale networks of the primate cortex are organized in a nested structure, whereby those responsible for sensorimotor processing are characterized by predominantly short-range interactions with nearby areas, and those linked to progressively more abstract or multimodal processing show a progressively larger proportion of connections with distant areas. Moreover, in this model areas that are unique to the primate brain would be expected to show a predominance of long-range connections[30]. To test this hypothesis, we quantified the ratio of labeled neurons representing local to distant projections in areas identified as belonging to different functional networks in the marmoset brain.

In this analysis, local connections were defined as those originating in neurons within 4 mm of the injection sites (a value sufficient to comprise all intrinsic connections, as well as projections from adjacent areas), whereas distant connections were those originating in cells located further than 8 mm from the injection sites. We conducted this analysis using areas corresponding to five functional networks (Fig. 7a) identified in the marmoset by neuroimaging experiments[45,48] and by correlation of data available in http://marmosetbrain.org with human neuroimaging[47]. The main hypothesis was upheld (Fig. 7b): the ratio of local to distant connections decreased from the primary sensorimotor network (Pri), to a network formed by interconnected higher-order sensorimotor and premotor areas (HOSom), to a network formed by frontal and posterior parietal visuomotor integration areas (VisM), and finally to putative homologs of the human default mode network (cognitive control network, CON[45], and apex transmodal network, APEX[47]). APEX and CON have been hypothesized to correspond to components of the human default mode networks (DMNs A and B, respectively[48]). There was no significant difference between the mean ratio of local to distant connections between CON/DMN-B and APEX/DMN-A, although areas comprising the latter tended to be more homogeneous in their connectivity ratio. This was the case even considering injections located relatively close to a primary sensory area (e.g. those in area TPO, which is located ~2 mm from the borders of the auditory core).

## Discussion

We report on a resource for quantitative analysis of cortico-cortical connectivity in a non-human primate (the marmoset monkey), made available to the scientific community via an online portal (http://marmosetbrain.org). The resource is unique in several aspects compared to others presently available for the non-human primate brain. First, the data are provided both in an aggregated form (using traditional formats, such as FLNe and SLN matrices linked to cytoarchitectural areas) and according to the results of each experiment (via three-dimensional and unfolded views of the brain, as well as downloadable files). Second, it provides access to entire datasets through an online interface that allows the user to judge the location of each labeled cell relative to the cortical cytoarchitecture (areas and layers), as well as to assess the quality of the underlying materials at points of interest. Third, the results have been registered to a common template of the marmoset brain using a computational pipeline which facilitates immediate comparison of results obtained in different individuals, as well as combined analyses involving many cases. This pipeline is distinctive in that it includes defined steps for expert validation of the registration results, aimed at ensuring that the attribution of cells to areas was as accurate as possible, based on current knowledge about the histological parcellation of the marmoset cortex. In our experience, this careful curation is necessary when dealing with data obtained in a genetically heterogeneous population of non-human primate brains, particularly in order to minimize false positive reporting of connections due to registration errors. The estimation of distances based on biologically realistic assumptions (simulated white matter tracts) also provides a significant improvement over previous approaches based on Euclidean or geodesic distances across the cortical surface. The potential of the resource as a platform for discoveries has been already demonstrated by recent studies based on mining data on the spatial distribution of labeled neurons[21,47,48] and analyses of network properties[49]. The present release significantly expands this capacity.

In building the resource, we focused on retrograde tracers, which allow visualization of individual cell bodies relative to

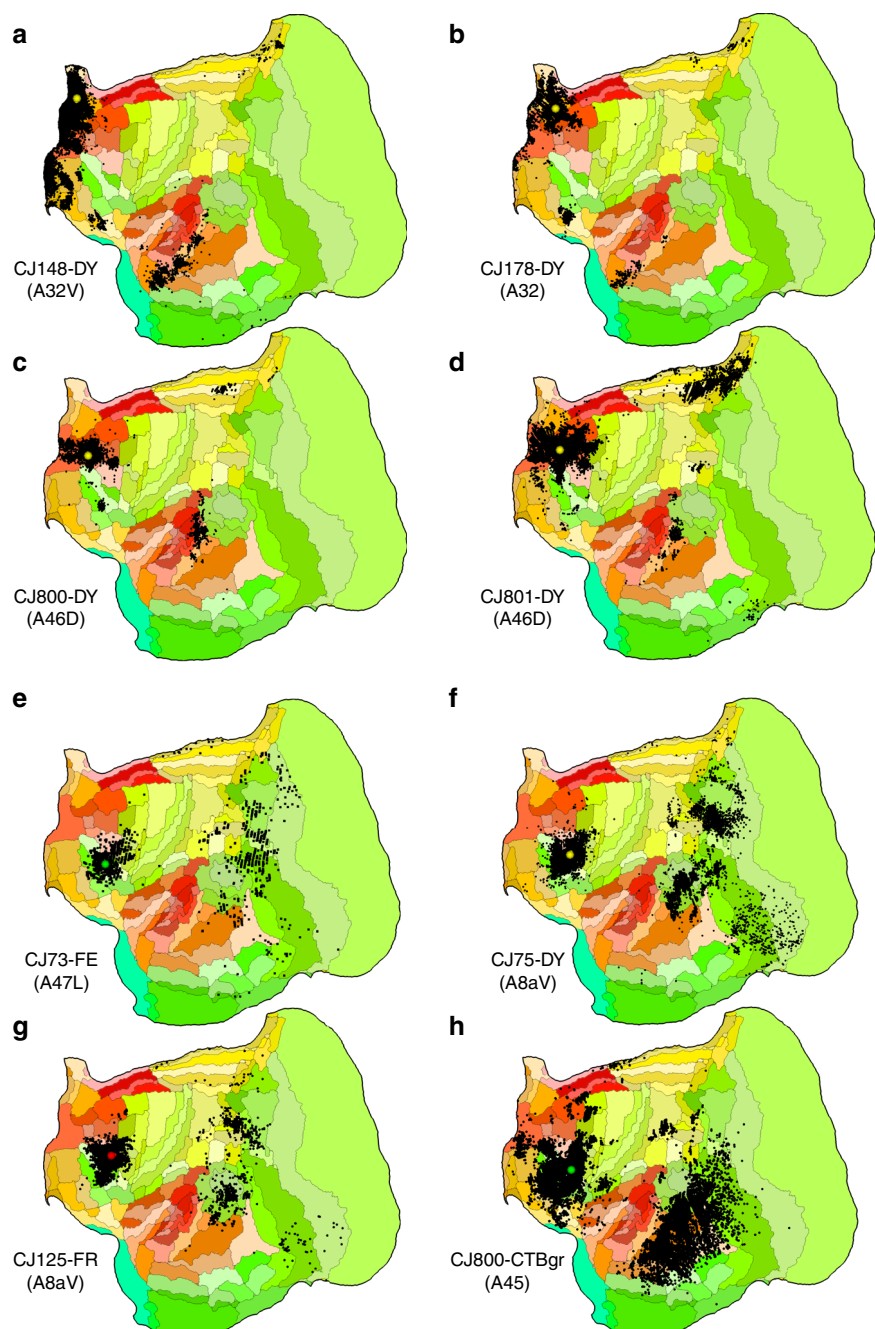

**Fig. 6 Patterns of labeled neurons following eight injections in prefrontal cortex.** These are visualized in flat reconstructions available through the portal. **a**, **b** Results of retrograde tracer injection in medial prefrontal areas (A32V: area 32 ventral; A32: area 32). **c**, **d** Injections in rostral dorsolateral prefrontal areas (A46D: area 46 dorsal); **e–h** injections in caudal dorsolateral and ventrolateral prefrontal areas (A47L: area 47 lateral; A8aV: area 8a ventral; A45: area 45). Even though long-range connections are a characteristic of all prefrontal areas, the medial and rostral areas (**a–d**) receive a higher proportion of their afferents from other prefrontal areas, in comparison with caudal areas (**e–h**). Source data are provided as a Source Data file.

cortical layers. Thus, each cell location shown in http://marmosetbrain.org is akin to a quantum of connectivity, an approach that enables accurate quantification with current technologies. This approach, which is in line with earlier resources for the macaque brain[16], is scalable to a complete cortical connectome, including information about the reciprocity of connections, based on future experiments to encompass the entire cortex. In comparison, the use of anterograde tracers imposes significant challenges for quantification, related to visualization and identification of synaptic terminals, and to distinguish those from axonal fragments on the way to their destinations. A similar resource incorporating anterograde data may require future development of techniques for automated identification and quantification based on high-resolution images (e.g. those available at http://marmoset.brainarchitecture.org). Some limitations remain regarding the quantification of labeled neurons near the injection sites, where it is impossible to distinguish cells that formed projections from those which may have incorporated the tracer by passive diffusion. Here, our approach was conservative, and most likely resulted in under-estimation of the actual numbers of intrinsic connections.

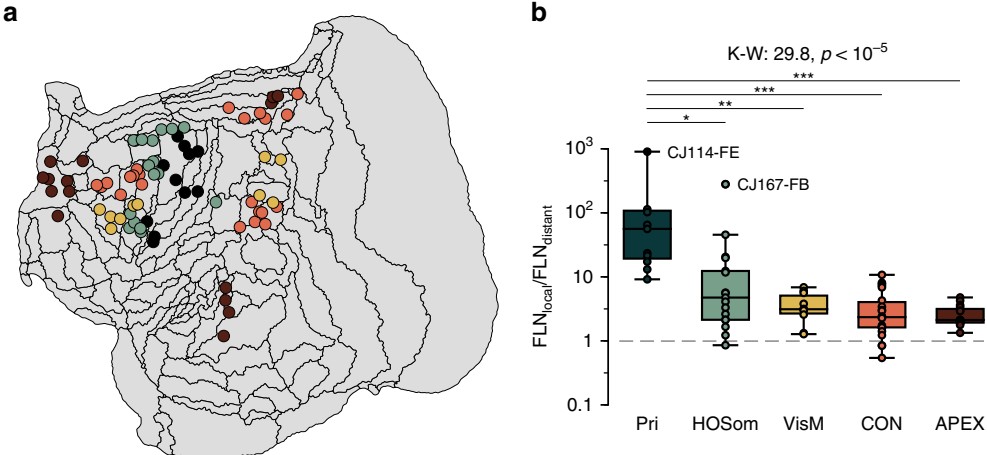

**Fig. 7 Local versus distant connectivity in cortical networks. a** Flat reconstruction of the marmoset cortex, showing the location of injection sites attributed to five networks of areas, defined in previous studies[45,47,48]. Pri (dark green): primary sensorimotor network (areas 3a, 3b, 4ab, and 4c); HOSom (light green): a network formed by interconnected higher-order somatosensory and premotor areas (dorsocaudal premotor area [6DC], medial premotor area [6M], caudal ventral premotor area [6Va], dorsal cingulate motor area [24d], and parietal area PF); VisM (yellow): a network formed by higher-order visuomotor integration areas (lateral and medial intraparietal areas [LIP and VIP], caudal subdivision of PE [PEc], and putative subdivisions of the frontal eye field [8aV and 8C]); CON (orange): cognitive control network (putative homolog of the default mode network B, including frontal areas 8aD and 6DR, cingulate areas PGM and 23b, and ventral parietal areas OPt and PG); APEX (brown): apex transmodal network (putative homolog of default mode network A, including areas 10, 23a, TPO, PGa/IPa, and the rostral part of TE3, near the temporal pole). **b** Box plot (center line: median; box limits: upper and lower quartiles; whiskers: 1.5× interquartile range; annotated points: outliers) of the ratio of the fraction of labeled neurons located within 4 mm of the center of the injection site (local connections) to neurons located >8 mm from the site (distant connections) following injections in the five networks (K–W: Kruskal–Wallis $H$ test: 29.8, p < $10^{-5}$). Statistical significance according to post-hoc Dunn's test: $p \leq 0.05$ (*), $p \leq 10^{-2}$ (**), $p \leq 10^{-3}$ (***). Source data are provided as a Source Data file.

One of the thorniest issues in neuroanatomy is the assignment of a given location to a specific cortical area. Areas with sharp, unambiguous histological boundaries are a minority, particularly among the association regions of frontal, parietal, and temporal cortex which experienced marked expansion in primate brain evolution[24]. This issue is not specific to the marmoset brain, as evident from the fact that widely used parcellations of the macaque cortex differ[16,25,26] and that estimates of the number of areas in the human brain have continued to increase[50]. Thus, unambiguous assignment of an injection site, labeled neuron, recording site, or activated voxel to an area is inherently uncertain. A tangible example of this problem relates to the quantification of the patterns of connectivity using matrices (e.g. Fig. 3). Such matrices provide convenient summaries of the data which are amenable to analyses (for example, based on graph theory), but should be interpreted with the biological reality of uncertain borders in mind. In the present resource, labeled cells and injections are assigned to specific cortical areas based on registration to the reference atlas[29], using a pipeline that ensured that clear cytoarchitectural boundaries were accurately represented (see section "Methods", "Expert-assisted registration to cortical areas"), but the definition of many other areas is less certain. In cases where the registration suggested that an injection involved more than one area, we assigned it according to the location of the injection barycenter. This decision was based on the fact that changes in the patterns of cortico-cortical connections do not occur abruptly at estimated borders, with gradual transitions of connectivity patterns often providing a better description[23,24,47,51,52], and that variations of patterns of connections also exist within single cytoarchitectural areas. Thus, excluding the data from injections that nominally crossed borders would imply a false level of precision, while resulting in selective reporting. Estimates of the percentage of injected voxels assigned to each area (Supplementary Table 3) and of the volumes of cortical areas[1] are both available, enabling estimation of the effect

of injections across borders[6] and statistical modeling based on multiple injections grouped in different ways[53].

The most important current limitation of the present resource is that it does not yet cover several groups of areas (e.g. those in the perirhinal, parahippocampal, insular, and anterior cingulate regions). This is being partially addressed by ongoing experiments which will be gradually released through the portal, but a more permanent solution may lie in a federated approach, whereby contributions from different laboratories can be integrated. Towards this end, the source code of both components of the Marmoset Brain Connectivity Atlas is released under the GPL license (see "Code availability" section), and the present injections are indexed through other online resources (e.g. http://marmoset.brainarchitecture.org, and http://marmosetbrainmapping.org)[2,54]. Another limitation is linked to the realities of experimental neuroanatomy: the quality of the materials is variable due to factors, such as unintentional damage to the tissue during surgery or histological processing or sub-optimal staining of histological sections, which may limit the interpretation of data in specific regions. We made a decision to show the material in its entirety, "warts and all", while including comments about known issues. We hope that this will allow users to make informed judgments on the suitability of specific materials to the analyses in mind while avoiding issues arising from selective reporting. The adoption of additional automation steps in the histological processing[2] will likely help address this limitation.

One of the desirable features going beyond the integration of additional materials would be the incorporation of probabilistic mapping of cortical areas, based on a template that reflects individual variability[55]. Presently the data are displayed and analyzed relative to the cytoarchitectural areas of a single brain examined in great detail[29], an approach which has limitations[1]. In developing the resource we have incorporated steps where the results of the registration were subject to validation by a human expert, but this is admittedly not as readily scalable as other steps.

Registration of the data to an average template of the brain, which incorporates probabilistic assignment of cortical areas to voxels, will facilitate principled analyses of the data collected through a fully automated pipeline, at least in terms of assignment of confidence intervals to inferences made based on the data. Another important development will be better integration of the datasets released through the present resource with those originating from magnetic resonance imaging, including diffusion imaging[54] and resting-state connectivity[39]. The precision of the non-invasive approaches for inference of neural connectivity has been questioned in terms of both sensitivity and selectivity[56–58]. Direct comparisons with the extensive ground truth data released through http://marmosetbrain.org provide a unique opportunity both to refine algorithms for MR image analysis, and to understand its limitations.

A position paper[5] which, in many ways, set the scene for the cellular-scale connectomic efforts of the present decade, proposed that this type of work should be based on experimental methods that are well-characterized, with individual steps for sample preparation, injection, histology, detection, and data analysis being stereotyped. The data collected from such an effort must be made freely available to all researchers from a centralized data repository, including raw image data, processed summary data, and metadata. The present resource provides the first large-scale platform for a non-human primate brain that allows quantitative analyses based on these principles. It will hopefully provide a fertile ground for future studies involving cellular connectivity, including comparative analyses with the comprehensive datasets being generated for the mouse and human brain using different techniques.

## Methods

**Surgical procedures.** Release 1.0 of the Marmoset Brain Connectivity Atlas (http://marmosetbrain.org) includes the results of 143 injections of retrograde tracers in 52 young adult (1.4–4.6 years, median age: 2.5 years) marmosets (31 male, 21 female). Detailed metadata for each animal is available via the Metadata panel on the portal. All experiments conformed to the Australian Code of Practice for the Care and Use of Animals for Scientific Purposes, and were approved by the Monash University Animal Experimentation Ethics Committee[19–21,41]. Intramuscular (i.m.) injections of atropine (0.2 mg kg$^{-1}$) and diazepam (2 mg kg$^{-1}$) were administered as pre-medication, before each animal was anaesthetized with alfaxalone (10 mg kg$^{-1}$, i.m.) 30 min later. Dexamethasone (0.3 mg kg$^{-1}$, i.m.) and amoxicillin (50 mg kg$^{-1}$, i.m.) were also administered prior to positioning the animals in a stereotaxic frame. Body temperature, heart rate, and blood oxygenation (pO$_2$) were continually monitored during surgery, and when necessary, supplemental doses of anesthetic were administered to maintain areflexia. Small incisions of the dura mater were made over the intended injection sites.

Six types of fluorescent tracers were used (Supplementary Tables 2 and 3, Supplementary Figs. 1 and 2): fluororuby (FR; dextran-conjugated tetramethylrhodamine, molecular weight 10,000, 15% in dH$_2$O), fluoroemerald (FE; dextran-conjugated fluorescein, molecular weight 10,000, 15% in dH$_2$O), fast blue (FB, 2% in dH$_2$O), diamidino yellow (DY, 2% in dH$_2$O), and cholera toxin subunit B (CTB, conjugated with either Alexa 488 [CTBgr] or Alexa 594 [CTBr], 1% in PBS). The dextran tracers resulted in bidirectional transport, but only retrograde labeling is reported here. The tracers were injected using 25 μl constant rate microsyringes (Hamilton, Reno, NV) fitted with a fine glass micropipette tip. Each tracer was injected over 15–20 min, with small deposits of tracer made at different depths. Following the last deposit, the pipette was left in place for 3–5 min to minimize tracer reflux. After the injections, the surface of the brain was covered with moistened ophthalmic film, over which the dural flaps were carefully arranged. The excised bone fragment was repositioned and secured in place with dental acrylic, and the wound closed in anatomical layers. Postoperative injectable analgesics were administered immediately after the animal exhibited spontaneous movements (Temgesic 0.01 mg kg$^{-1}$, i.m., and Carprofen 4 mg kg$^{-1}$, s.c.), followed by oral Metacam (0.05 mg kg$^{-1}$) for 3 consecutive days.

**Histological processing.** Survival times varied between 3 and 22 days (median: 15 days), after which the animals were anesthetized with alfaxalone (10 mg ml$^{-1}$ i.m.) and, following loss of consciousness, administered an overdose of sodium pentobarbitone (100 mg kg$^{-1}$, i.v.). They were then immediately perfused through the heart with 500 ml of heparinized saline, followed by 500 ml of 4% paraformaldehyde (PFA) in 0.1 M phosphate buffered saline (PBS; pH 7.4). The brains were post-fixed in the same medium for at least 24 h, and then immersed in buffered PFA with increasing concentrations of sucrose (10–30%). They were then sectioned (40 μm thickness) in

the coronal (most cases) or parasagittal (three hemispheres) plane, using a cryostat. One section in five was mounted unstained for examination of fluorescent tracers, and coverslipped after quick dehydration (2 × 100% ethanol) and defatting (2 × xylene). Adjacent sections were stained for cell bodies (using the cresyl violet stain, or the NeuN stain[59]), cytochrome oxidase, or myelin (for protocols, see Supplementary Table 2). The remaining section in each series was stored in cryoprotectant solution in a freezer, to be used as a backup in the case of unsatisfactory staining or damage during the processing of the histological sections. Hence, the spacing between adjacent sections in each series was 200 μm. Stained sections were scanned using an Aperio Scanscope AT Turbo system (Leica Biosystems), providing a resolution of 0.50 μm pixel$^{-1}$. For cases with injections in the left hemisphere, the resulting images were flipped horizontally to preserve the common format of all cases and to facilitate comparisons.

**Microscopic analysis and digitalization.** Sections were examined using epifluorescence microscopes. Labeled neurons were identified using ×10 or ×20 dry objectives, and their locations within the cortex and subcortical structures were mapped using a digitizing system attached to the microscope. To minimize the problem of overestimating the number of neurons due to inclusion of cytoplasmic fragments, labeled cells were accepted as valid only if a nucleus could be discerned. This was straightforward in the case of DY, since this tracer only labels the neuron's nucleus[60]. In the case of tracers that label the cytoplasm (FB, FE, FR, CTBg, and CTBr), the nucleus was discerned as a profile in the center of a brightly lit, well-defined cell body, which in the vast majority of cases had an unmistakable pyramidal morphology.

**Reference brain atlas.** We selected the Paxinos et al.[29] stereotaxic atlas of the marmoset brain as the three-dimensional reference space. The system of coordinates in this atlas is based on cranial landmarks: the horizontal zero plane is defined as the plane passing through the lower margin of the orbits and the center of the external auditory meatuses, the anteroposterior zero plane is defined as the plane perpendicular to the horizontal zero plane which passes the centers of the external auditory meatuses, and the left–right zero plane is the midsagittal plane. The Nissl-stained plates illustrated in the PDF edition of the book (available at http://marmosetbrain.org/reference) were converted into a 3D template[1,61]. This resulted in a volumetric image of a resolution of 40 × 500 × 40 μm (mediolateral, rostrocaudal, and dorsoventral, respectively) which preserves the stereotaxic coordinates of the source atlas plates. The atlas parcellation scheme consists of 116 cortical areas arranged in an ontology comprising 22 groups. The same scheme has been also adopted by other large-scale projects, such as BRAIN/Minds[62] and the NIH Marmoset Brain Atlas[54], ensuring present-day interoperability.

**3D reconstruction and mapping procedure.** To register the results of individual experiments into the reference template in a systematic and standardized way a computational workflow was established. The pipeline[1] computes a set of spatial transformations that allow expressing the location of any object of interest in the experimental data, such as labeled cell body or location of the injection site, within a common set of stereotaxic coordinates[29]. This process enables analyses of the connectivity patterns obtained in different cases using a common spatial and semantic reference. Three main steps can be distinguished: The affine reconstruction step, the deformable reconstruction, and the combined, affine and deformable, co-registration with the template image (Supplementary Fig. 5C–E). The pipeline is based on the Possum 3D reconstruction framework[63] (https://github.com/pmajka/poSSum) and the Advanced Normalization Tools (ANTS) software suite[64] (http://picsl.upenn.edu/software/ants/).

First, the images of the Nissl-stained sections were arranged in rostrocaudal order and edited so that only parts representing the brain tissue of the hemisphere to be reconstructed were preserved. The digitized locations of the labeled cells plotted on the fluorescent series were then aligned to adjacent Nissl sections (Supplementary Fig. 5A). For the purpose of the 3D reconstruction, the high-resolution images of Nissl-stained sections are downsampled to 40 μm × 40 μm resolution to reduce the processing time without significantly hampering the mapping accuracy. The affine reconstruction step (Supplementary Fig. 5C) recovered the overall anatomical shape of an individual's brain hemisphere. This was achieved by two alternating procedures: 3D registration of the reconstruction to the template, and 2D rigid alignment of the sections' images to each other[1,63]. This step restored the general shape of a hemisphere, but did not address noticeable transitions between consecutive sections caused by tissue distortions, which naturally occur during the histological processing. The deformable reconstruction step (Supplementary Fig. 5D) reduced these distortions by deformably coregistering an image of a given section to a synthetic image obtained by averaging the images of neighboring sections[63]. This process produced a smooth reconstruction with a more natural appearance, which also facilitated the subsequent registration to the reference template.

In the last step, the reconstruction of a hemisphere was brought into the space of the reference template (Supplementary Fig. 5E, F). This procedure mitigated the differences between the experimental and the template hemispheres by morphing the reconstruction to match the reference image, and minimized the influence of artifactual distortions, such as deformation of the surface due to the surgical

procedures, or damage incurred during the extraction of the brain from the skull. Both the 3D reconstructions and the atlas image were resampled to an isotropic resolution of 75 μm and smoothed with a median filter with a 1 voxel radius. During the affine step, the coregistration was driven by the Mattes mutual information[65] (MI) image similarity metric. For the deformable registration two, equally weighted, metrics were used: the cross-correlation coefficient[66] (CC) with a kernel size of 5 voxels, and the Point-Set Expectation[64] (PSE) metric, which forced corresponding label maps to overlap (see below). Overall, this process established a precise mapping between the experimental dataset and the template, which allows one to map the sections of a specific animal into the stereotaxic space.

**Expert-assisted registration to cortical areas**. To further increase the accuracy of the mapping (and, consequently, the assignment of the individual labeled cells to cortical areas) an additional processing step was introduced to the workflow initially described[1]. This procedure relied on guiding the registration to maximize the overlap of corresponding cytoarchitectural areas across individual experimental brains and the template brain.

Two sets of label maps were systematically outlined in each case, with each set given equal weight in driving the registration. The first set (Supplementary Fig. 6A, B) was drawn on images of Nissl-stained sections downsampled to 40 μm resolution, and highlighted the major morphological features of the brain, such as the ventral and dorsal banks and the fundus of the calcarine sulcus, lateral and medial banks and the fundus of the lateral sulcus, hippocampus, lateral geniculate nucleus, and claustrum. In addition, in the majority of the cases (45 out of 53), the outline of the entire cortex was delineated, thus ensuring that prominent morphological landmarks, such as the frontal, occipital, and temporal poles were precisely captured.

The second collection of label maps corresponded to the borders of individual cortical areas or aggregates of areas (Supplementary Fig. 6C). These were traced on the set of sections used in the reconstruction by an experienced neuroanatomist (M.G.P.R.), using the information available from the entire set of histological sections for an animal (that is, the Nissl/NeuN, myelin, and cytochrome oxidase series). The process of establishing which areas were to be included was carried out iteratively for each case in the following way: (1) the outline of the cortex and the gross morphological landmarks were outlined, (2) the coregistration was carried out using a small number of histological areas, and the results were inspected, and (3) depending on the outcome, additional areas were added, and steps 2 and 3 were repeated until satisfactory results (i.e., no obvious assignment of labeled cells to incorrect areas, as determined by histology) were achieved.

In most cases, the registration was based on 14 histological labels distributed across the cortex, which included: The primary visual (V1) and somatosensory (A3b) areas, the middle temporal area (V5/MT), the combined extent of auditory core and belt areas (Aud), area prostriata (ProSt), the entorhinal cortex (Ent), the piriform cortex (Pir), parahippocampal area TH, the area 13 complex in orbitofrontal cortex, the area 3 complex in posterior cingulate cortex, medial prefrontal area 32V, dorsolateral prefrontal area 8b, the lateral intraparietal area (LIP), and temporal area PGa/IPa. Each of these areas, or groups of areas, could be unambiguously defined with high precision on the basis of having sharp architectural borders. In many animals, additional areas were used to drive the registration based on factors including markedly different sulcal morphology in regions of interest containing labeled neurons or injection sites, and persisting errors in the assignment of cells to areas which were obvious during successive manual inspection steps. The actual number of areas used for this purpose varied between 2 and 22 in different cases, depending on the accuracy obtained with successive registration steps (Supplementary Fig. 6C).

**Quantification of connectivity patterns**. The locations of cells and injection sites were mapped into the template using the computed set of spatial transformations (Supplementary Fig. 5G). Subsequently, they were assigned to a cortical area (or adjacent areas; see Supplementary Table 3) based on the stereotaxic location of the corresponding voxel(s) relative to the atlas parcellation. As indicated in Supplementary Table 3, 79 injection sites were entirely contained within estimates of a single cytoarchitectural area, 41 were >80% contained within an area (i.e. well within the estimated precision of the method used to reconstruct boundaries), and 23 likely crossed borders. The assignment to an area was based on the voxel containing the injection barycenter, and validated by an expert. Estimated percentages of the voxels in different areas comprised by the injection sites are given in Supplementary Table 3.

In addition to being assigned to an area, labeled cells were divided into either those above layer 4 (supragranular) or below it (infragranular) by a procedure that involved manual delineation of the granular cell layer across the entire set of sections that contained labeled neurons (Supplementary Fig. 5I). Cells located in entorhinal cortex (Ent), piriform cortex (Pir), amygdalopiriform transition area (APir), and area 29a-c (A29a-c) were excluded from this assignment due to the lack of a visible layer 4. In the intermediate and medial sectors of the primary motor cortex (A4ab) and dorsocaudal premotor area (A6DC) the interface between layers 3 and 5 was used for this purpose. The 3D locations, assigned areas, and laminar positions were stored in a database. Based on these data, the connectivity patterns obtained for each injection were quantified in a way similar to the one used in other studies involving injections of retrograde tracers[15] to assure compatibility and to enable cross-study comparisons.

Specifically, the strength of a directed connection to an injected area $A$ from an area $B$ was defined as the Fraction of Labeled neurons (extrinsic; FLNe), Eq. (1),

$$\text{FLNe}_{B \to A} = \frac{\text{Number of neurons projecting to area } A \text{ from area } B}{\text{Total number of neurons projecting to area } A \text{ from all areas } - \text{ neurons identified in area } A}$$

(1)

which yields values ranging from 0 to 1 for each projection. The within area connections (i.e. from area $A$ to itself) weights were not considered. Furthermore, based on the established position with respect to the layer 4, the fraction of neurons which originate in the supragranular layers (supragranular neurons, SLN) of the source area was obtained according to Eq. (2).

$$\text{SLN}_{B \to A} = \frac{\text{Number of supragranular neurons projecting to area } A \text{ from area } B}{\text{Total number of neurons projecting to area } A \text{ from area } B}$$

(2)

The FLNe and SLN values were calculated for each injection separately. For areas where multiple injections were placed, an average FLNe value was obtained using the arithmetic mean. A corresponding average SLN value was obtained by first summing supragranular and infragranular neurons across injections in which a specific projection was observed, followed by calculating the SLN off the obtained totals[15].

**Estimation of interareal distances**. Distances between cortical areas (interareal distances), as well as the distances between individual cells and injection sites, were estimated by measuring the lengths of simulated axonal tracts connecting these areas (Supplementary Fig. 3). The procedure stemmed from the assumption that, at the mesoscopic level, the brain organization follows the arrangement that minimizes the wiring length, or more general—the wiring cost[67,68]. Therefore, we developed a computational model which assumes that hypothetical axonal bundles linking any two given points in the cortex minimize the wiring length by following the fastest (the least expensive, geodesic) possible trajectory through the white matter. This was accomplished numerically by assigning to each voxel of the 3D brain template a scalar parameter which expresses the ease with which a simulated fiber would traverse the said voxel. Fibers can pass through the white matter easily ($f_{\text{WM}} = 1.0$), while traveling across gray matter is difficult, yet still possible ($f_{\text{GM}} = 0.05$). On the other hand, reaching outside the brain is prohibited: $f_{\text{OUT}} = 0$.

To calculate the interareal distance matrix (Supplementary Fig. 3D), for each cortical area its centroid (defined as the maximum of the signed distance transform applied to the binary mask of a given area) was established. Subsequently, three-dimensional geodesic trajectories connecting each pair of centroids were calculated using the Fast Marching[69] method implemented in the SimpleITK framework (https://itk.org/Doxygen/html/classitk_1_1FastMarchingImageFilter.html) taking into account the imposed constraints. To mitigate numerical errors, for each pair of areas trajectories in both directions were calculated (e.g. for areas MT and V2, the paths from MT to V2 as well as from V2 to MT were obtained). The length of each trajectory was measured, and the distance between areas $i$ and $j$ ($d_{ij}$) was defined as the average length of paths in both directions, Eq. (3).

$$d_{ij} = \frac{1}{2}(d_{i \to j} + d_{i \leftarrow j})$$

(3)

The simulated tracts appear biologically plausible[70] (Supplementary Fig. 3A, B). For instance, those connecting areas in close proximity are superficial to the interface between gray and white matter and form characteristic U-shaped trajectories, while those linking remote areas pass through deeper parts of the white matter. The trajectories in the opposing directions (i.e. from area $i$ to $j$ and from $j$ to $i$) varied little in the vast majority of cases. The discrepancy (Supplementary Fig. 3C) was ≤0.046 for 95% of the pairs of areas and ≤0.129 for 99% of the cases which illustrates the numerical stability of the method. Sporadically, however, due to the relatively large spacing between consecutive atlas plates (500 μm), some numerical errors occurred causing trajectories in opposing directions to differ noticeably. Specifically, 15 pairs of areas have discrepancy larger than 0.3, the majority of which located in parts of the brain where the morphology changed rapidly in comparison with the spacing of the atlas plates. The interareal wiring distance matrix (Supplementary Fig. 3D) comprises 6670 values ($n(n - 1)/2$, where $n$ is the number of cortical areas), distributed unimodally (Supplementary Fig. 3E). The average interareal distance of 12.5 mm is biologically plausible, comparable to the half of rostrocaudal extent of the marmoset brain (~30 mm).

Finally, to eliminate bias resulting from choosing a specific relation between the ease of passing through the gray or white matter ($f_{\text{WM}}/f_{\text{GM}} = 20$) the analyses were repeated for $f_{\text{WM}}/f_{\text{GM}}$ ratios ranging from 30 (a substantial difference between the gray and white matter) to 1 (tissue classes indistinguishable). The results were largely independent for $f_{\text{WM}}/f_{\text{GM}}$ ratios within a range that appears biologically relevant (10–30). Only when the difference between the tissue classes started effacing, the average interareal distance dropped sharply (Supplementary Fig. 3F).

**Online portal**. The primary experimental data, together with the results of the computational workflow described above, have been made open to the public by the means of the http://marmosetbrain.org portal. Supplementary Fig. 7 provides an abridged overview of the portal architecture, supporting infrastructure, and its relation to the computational pipeline described above.

The results generated by the data-processing pipeline (Supplementary Fig. 7, bottom left) are of a heterogeneous nature, including images, text, and tables, which calls for dedicated ways of handling each type in order to allow online sharing. The voluminous high-resolution images of histological sections are converted to JPEG 2000 format and transferred to the datastore. To facilitate responsive and fluent user experience when displaying the images online, they are handled by a dedicated image server (Djatoka, https://sourceforge.net/projects/djatoka/) equipped with an elaborate caching and streaming mechanism to promptly access regions of interest at various magnifications.

Text and tabular data are stored in a master database (Supplementary Fig. 7, bottom right) and subsequently refined into static datasets, such as spreadsheets with quantified connectivity patterns, segmentation of the sections into cortical areas, compiled sets of metadata, static visualizations (e.g. spatial patterns of labeled cells presented as 3D images), etc. These datasets are made instantly available to the web server, eliminating the necessity of generating them upon user's request. Since the results are computed offline and only once throughout the entire infrastructure, the possibility of a discrepancy is ruled out and the chances of a failed request for obtaining a specific result are reduced. This arrangement noticeably simplifies continuous updates, expansion, maintenance, and curation of the connectivity atlas.

From the user's perspective, the http://marmosetbrain.org portal consists of two components (Supplementary Fig. 7, top). The high-resolution section viewer employs the OpenLayers (https://openlayers.org/) framework to combine the high-resolution images with the non-imaging data, providing views of the primary experimental data. The complementary analytics interface provides the capability of exploring the quantitative results of the tracer injection experiments. Each component can be accessed either using a web browser or via a dedicated application programming interface (API), both offering access to the same range of data.

**Reporting summary**. Further information on research design is available in the Nature Research Reporting Summary linked to this article.

## Data availability

The cortico-cortical connectivity datasets (RRID:SCR_015964) generated and analyzed in the current study are available under the terms of Creative Commons Attribution-ShareAlike 4.0 License and publicly available through the Marmoset Brain Connectivity Atlas portal (http://marmosetbrain.org). The data underlying Figs. 5–7, as well as Supplementary Figs. 1–4, and 6 are provided as a Source Data file. A reporting summary for this Article is available as a Supplementary Information file.

## Code availability

The code for reproducing the analyses presented in Figs. 5–7, Supplementary Figs. 1–4, and 6 is provided in supplementary materials. The source code of both components of the http://marmosetbrain.org is released under the GPL license (see: https://github.com/Neuroinflab/marmosetbrain.org and https://github.com/Neuroinflab/analysis.marmosetbrain.org).

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

## Acknowledgements

The authors would like to acknowledge the efforts of many other individuals who spent many hours examining the experimental materials under the microscope, conducting digital image analysis, or developing software components, which made this resource possible, including Dr. Nafiseh Atapour, Dr. Kathleen Burman, Lorenzo Canalini, Dr. Michela Gamberini, Dr. Daniele Impieri, Daria Malamanova, Susan Palmer, Jan Mąka, Karyn Richardson, Maciej Śmigielski, Gabriela Saworska, Ianina H. Wolkowicz, Amanda Worthy and Sherry Zhao, Technical support from Kirsty Watkins in conducting histological processing, and the contributions of Dr. Tristan Chaplin and Dr. Hsin-Hao Yu to early phases of the development of the methods are also gratefully acknowledged. This work utilized the Multi-modal Australian ScienceS Imaging and Visualization Environment (MASSIVE) high-performance computing infrastructure (https://www.massive.org.au/), and scanning of histological slides was performed by the Monash Histology Platform (https://platforms.monash.edu/histology/). Funding for the experiments and for the development of the resource was provided by the Australian Research Council (DP110101200, DP140101968, CE140100007), the National Health and Medical Research Council (APP1003906, APP1020839, APP1083152, APP1082144, APP1122220), the International Neuroinformatics Coordinating Facility (INCF Seed Funding grant scheme), the European Regional Development Fund under the Operational Program Innovative Economy (POIG.02.03.00-00-003/09), the National Centre for Research and Development (ERA-NET-NEURON/17/2017), the National Institute of Mental Health (R01MH062349), the Office of Naval Research (N00014-17-1-2041), Cold Spring Harbor Laboratory (Crick-Clay Professorship), the Mathers Foundation, and the Indian Institute of Technology Madras (H. N. Mahabala Chair).

## Author contributions

Conceptualization: M.P.G.R., P.P.M.; Methodology: P.M., P.P.M., M.P.G.R., D.K.W.; Software: P.M., S. Bai, S. Bednarek; Validation: P.M., M.P.G.R.; Data Analysis: P.M., S. Bakola, J.M.C., N.J., L.P., P.T., K.H.W., D.H.R., M.P.G.R.; Data Curation: P.M., M.P.G.R.; Writing: P.M., M.P.G.R.; Review and Editing: all authors; Visualization: P.M., S. Bai; Supervision: P.M., P.P.M., X.-J.W., D.K.W., M.P.G.R.; Project Administration: P.M., P.P.M., X.-J.W., D.K.W., M.P.G.R.; Funding Acquisition: P.M., P.P.M., D.K.W., M.P.G.R.

## Competing interests

The authors declare no competing interests.
