## [Peer Review File · Nature Communications]

Reviewers' Comments:

Reviewer #1:

Remarks to the Author:

The present publication, the data it represents, and the open source tools for navigating the data are field defining.

To appreciate the significance of this work it is important to note that there is a long history of integrating monkey tract-tracing and histological data to understand brain systems, primarily in the macaque. The marmoset, sharing a more distant lineage with humans, is a small primate that have garnered tremendous recent interest because it is genetically accessible, causing many individual and national efforts to build understanding and new tools to study the unique non-human primate model. The present work brings together much of the world's tract tracing data into a common coordinate system and framework of open data analysis. This is a milestone.

There are multiple strengths to the manuscript in addition to its landmark status. First, it illustrates an excellent set of multi-scale data visualization tools that can toggle between visualization of histological sections (and labeled neurons) and the broad view enabled by the flat map representations. Second, the datasets are offered in survey forms that can be easily navigated, such as is possible when surveying across multiple injections in the flatmap summary. Finally, the data can be quantified as illustrated extensively by the test of the distance hypothesis. That is, in addition to be an interesting test of the hypothesis, the process of testing the hypothesis and the multiple analyses illustrate the broad utility of the effort.

The manuscript is in excellent form and could be published with minimal modifications. There are a few suggestions for augmentations that might be considered.

First, in the analysis of distance, it is unclear to what degree the presence of extensive local connections could mask the relative presence of widespread distant connections. That is, it might be useful to see some form of plot that separately quantifies some measure of local and some measure of distant connectivity separately so that one can directly see how the areas are changing. Also, it would be interesting to know a bit more about any anomalies, such as TPO. That is, can something be learned from the exceptions as much as from the general rule?

Second, while space is limited, the motivating hypotheses for many of the analyses (the Oligschlager papers) was incomplete in its description in the sense that the 2017 and 2019 papers were direct tests of the hypothesis put forth by Buckner and Krienen (2013), itself based on prior existing human and macaque connectivity distance estimates. Thus, while the present paper offers convergent evidence for the Buckner and Krienen hypothesis, and does so with a level of rigor not previously possible, the 2019 cited paper was not the origin of the hypothesis and not the first evidence.

Finally, of all the figures and analyses, the results of the simulated path lengths illustrated in Figure 5 was the least compelling, in part because it represents simulated anatomy.

Reviewer #2:

Remarks to the Author:

The manuscript introduces an online resource for anatomical data from the marmoset brain.

Online availability and high-quality interaction with large-scale anatomical data is an important goal

and a very helpful resource for the neurosciences. In particular, the study of species beyond mouse is of utmost importance to be able to put findings from mouse into evolutionary context, and to avoid over-fitting on one species.

I therefore think the presented resource is relevant and deserves visible publication.

I am however rather confused about the particular advance that the presented data/analysis methodology represents:

the authors have previously published about the same goals, in particular Lin et al 2019 elife. Already there, online resources are linked. The currently available online resource already shows data from the 143 brains that are advertised in this manuscript.

So unless I am simply confused about this, I think at the very least it should be made crystal clear to the reviewer/reader what has and what has not been published, and what exactly the advance of this manuscript is. If it is simply the description of an online browsing tool, I am less overwhelmed. If it includes the 143 tracing experiments (as introduction page 3 implies), it's more compelling. Or have these been published before?

In any case, the ability to represent and analyse each and every injection at high-resolution in 3D space, and the possibility to address co-variability of injection location with projection data is of high relevance, and provides the necessary level of detail to draw relevant conclusions from this kind of projection-level anatomical data. Has this aspect been published before or made available before?

In summary, if the attribution of data access and resource progress to the various publications can be made clear (maybe in a first figure panel), and there remains sufficient novelty mass assigned to this very manuscript, I consider this work of high importance and quality.

Reviewer #3:

Remarks to the Author:

 Overall, marmosetbrain.org has been (and will likely continue to be) a tremendously useful resource for the marmoset neuroscience community - the authors should be highly commended for making these vast amounts of valuable and very labour-intensive data open-source and providing an organized resource for accessing these data.

Major comments:

1. For the reader, all of the novel aspects of this manuscript may not be explicitly clear with reference to the technical paper already published by Majka et al., 2016. The last paragraph of the introduction states "Here we introduce an online resource which addresses the goals mentioned above. The Marmoset Brain Connectivity Atlas (www.marmosetbrain.org) provides interactive access to a large collection of data on the cortico-cortical connections, obtained using retrograde tracers." Wasn't this resource already introduced in Majka et al, 2016? In general, it would be helpful to explicitly state (and do so earlier in the manuscript) what the present manuscript adds to the aforementioned paper, which also directs readers to marmosetbrain.org. As it stands, this is not stated until the 3rd paragraph of the introduction that features were added and additional data was made open source. The additional work (e.g., refinements of the mapping) becomes clear in the methods section as well, but these improvements should be collated into a few sentences early on, rather than presented piecemeal throughout the manuscript.

2. Although the connectivity profiles in matrix format are somewhat useful for comparing to magnetic resonance imaging techniques (e.g., rsfMRI and DTI), allowing users to directly download the tracer data in 3D volume format (e.g., a single nifti file for each injection with connectivity strengths) in a template space would be much more useful and allow for ready comparisons to topologies acquired by MRI. I would strongly urge the authors to consider this possibility. Having a downloadable structural volume (e.g., like that provided with the NIH marmoset template) to register to would also aide in this effort.

Stefan Everling

Detailed reply to reviewers

Reviewer comments in italics, our replies in normal font

Reviewer #1

The present publication, the data it represents, and the open source tools for navigating the data are field defining.

To appreciate the significance of this work it is important to note that there is a long history of integrating monkey tract-tracing and histological data to understand brain systems, primarily in the macaque. The marmoset, sharing a more distant lineage with humans, is a small primate that have garnered tremendous recent interest because it is genetically accessible, causing many individual and national efforts to build understanding and new tools to study the unique non-human primate model. The present work brings together much of the world's tract tracing data into a common coordinate system and framework of open data analysis. This is a milestone.

There are multiple strengths to the manuscript in addition to its landmark status. First, it illustrates an excellent set of multi-scale data visualization tools that can toggle between visualization of histological sections (and labeled neurons) and the broad view enabled by the flat map representations. Second, the datasets are offered in survey forms that can be easily navigated, such as is possible when surveying across multiple injections in the flatmap summary. Finally, the data can be quantified as illustrated extensively by the test of the distance hypothesis. That is, in addition to be an interesting test of the hypothesis, the process of testing the hypothesis and the multiple analyses illustrate the broad utility of the effort.

The manuscript is in excellent form and could be published with minimal modifications. There are a few suggestions for augmentations that might be considered.

First, in the analysis of distance, it is unclear to what degree the presence of extensive local connections could mask the relative presence of widespread distant connections. That is, it might be useful to see some form of plot that separately quantifies some measure of local and some measure of distant connectivity separately so that one can directly see how the areas are changing. Also, it would be interesting to know a bit more about any anomalies, such as TPO.

In response to this suggestion, we have conducted a new analysis, where we quantify the ratio of local to distant connections in areas associated with different cortical resting state networks (RSNs). This analysis is now presented in Figure 7, and associated text in the last section of Results. This represents a quantitative, cellular-level analysis of the hypothesis advanced by Buckner and Krienen (2013), which postulates a gradual shift in the emphasis in local versus distant connectivity as one progresses from cortical networks that are evolutionarily conserved, towards those that are unique to the primate brain. Under this framework, area TPO does not appear as an outlier; it shows exactly the type of connectivity expected for a member of the "default mode network".

Second, while space is limited, the motivating hypotheses for many of the analyses (the Oligschlager papers) was incomplete in its description in the sense that the 2017 and 2019 papers were direct tests of the hypothesis put forth by Buckner and Krienen (2013), itself based on prior existing human and macaque connectivity distance estimates. Thus, while the present paper offers convergent evidence for the Buckner and Krienen hypothesis, and does so with a level of rigor not previously possible, the 2019 cited paper was not the origin of the hypothesis and not the first evidence.

The reviewer is correct, and we have edited several parts of the manuscript accordingly.

Finally, of all the figures and analyses, the results of the simulated path lengths illustrated in Figure 5 was the least compelling, in part because it represents simulated anatomy.

We have accepted this suggestion, and accordingly integrated Figure 5 of the previous version into one of the Supplementary figures (Supplementary Figure 3). Thus, the number of figures in the paper is maintained (7) despite the inclusion of the new analysis figure described above.

Reviewer #2

The manuscript introduces an online resource for anatomical data from the marmoset brain.

Online availability and high-quality interaction with large-scale anatomical data is an important goal and a very helpful resource for the neurosciences. In particular, the study of species beyond mouse is of utmost importance to be able to put findings from mouse into evolutionary context, and to avoid over-fitting on one species.

I therefore think the presented resource is relevant and deserves visible publication.

I am however rather confused about the particular advance that the presented data/analysis methodology represents:

the authors have previously published about the same goals, in particular Lin et al 2019 elife.

Already there, online resources are linked. The currently available online resource already shows data from the 143 brains that are advertised in this manuscript.

So unless I am simply confused about this, I think at the very least it should be made crystal clear to the reviewer/reader what has and what has not been published, and what exactly the advance of this manuscript is. If it is simply the description of an online browsing tool, I am less overwhelmed. If it includes the 143 tracing experiments (as introduction page 3 implies), it's more compelling. Or have these been published before?

There are two points that we would like to make in response to these comments.

The first, specific point, is that the present resource does represent a significant advance over previous efforts, which were the subject of recent papers by a subset of the present authors. We acknowledge that the previous version of the current manuscript was somewhat unclear in this respect, by mentioning specific advances in a piecemeal manner. We have addressed this by taking a suggestion made by reviewer 3, and reorganizing the Introduction in such a way that it immediately places the present paper in better context.

In summary, we have previously demonstrated the feasibility to integrate results of multiple individuals in a common template brain and to share entire histological datasets online, in a “toolbox” paper (Majka et al. 2016). The present resource takes advantage of these developments and represents the maturation of this process. It includes fully curated versions of the data, obtained by a new pipeline that improves accuracy, and these are for the first time integrated with an analysis interface that allows meaningful exploration of patterns of cortico-cortical connectivity. In addition, the new interface includes original data such the stereotaxic coordinates of labeled neurons, individual case and aggregated data visualizations, estimates of the distances between areas calculated using realistic assumptions, and facilities for integration with neuroimaging data, as well as new functionality for data mining.

The second point is more general, but worth highlighting as it represents an important consideration in this era of efforts when open-science platforms are considered a desirable “way forward” to accelerate discovery. This project was designed as an open resource, from start. Thus, we have always encouraged other online platforms to include links to our materials, even in preliminary form, provided adequate citation. This is an example of the “federated approach” we describe as desirable, in the Discussion of our paper. However, it would be a clear disincentive to openness if such practice were to be considered previous publication, with consequent implications for future open science projects. We have edited the Discussion of this paper to clarify the point that **the dataset that is reported in the present paper is the primary source**, to which other websites link.

Specifically, the paper by Lin et al. (2019) is focused on describing and validating the methodology developed for an independent project, which also aims to create a resource for visualization of connections in the marmoset brain. This resource is focused on original materials raised by the RIKEN/ Cold Spring Harbor Laboratory group, but includes links to earlier versions of the present materials. In the paper by Lin et al. (2019) the only mentions of our materials occur in the Introduction (“[For the marmoset, an online database of >140 retrograde tracer injection studies in about 50 cortical areas is available online \(http://monash.marmoset.brainarchitecture.org\)](http://monash.marmoset.brainarchitecture.org) (Majka et al., 2016)”, and Discussion (“In addition, analysis of injection centers show proximity/overlap of injections from a previous data set from the Rosa laboratory for which 3D spatial information is available (Appendix 10). This should permit virtually increasing N for this project”).

In any case, the ability to represent and analyse each and every injection at high-resolution in 3D space, and the possibility to address co-variability of injection location with projection data is of high relevance, and provides the necessary level of detail to draw relevant conclusions from this kind of projection-level anatomical data. Has this aspect been published before or made available before?

No, these aspects of the resource are being reported for the first time. We reiterate the fact that the resource we submit for publication is not yet in the public domain, but will be done so immediately upon acceptance of the manuscript.

In summary, if the attribution of data access and resource progress to the various publications can be made clear (maybe in a first figure panel), and there remains sufficient novelty mass assigned to this very manuscript, I consider this work of high importance and quality.

We hope that the explanation above has clarified the relationship between the present paper and those of Majka et al. (2016) and Lin et al. (2019).

Reviewer #3

Overall, marmosetbrain.org has been (and will likely continue to be) a tremendously useful resource for the marmoset neuroscience community - the authors should be highly commended for making these vast amounts of valuable and very labour-intensive data open-source and providing an organized resource for accessing these data.

Major comments:

1. For the reader, all of the novel aspects of this manuscript may not be explicitly clear with reference to the technical paper already published by Majka et al., 2016. The last paragraph of the introduction states "Here we introduce an online resource which addresses the goals mentioned above. The Marmoset Brain Connectivity Atlas (www.marmosetbrain.org) provides interactive access to a large collection of data on the cortico-cortical connections, obtained using retrograde tracers." Wasn't this resource already introduced in Majka et al, 2016?

The Majka et al. 2016 paper was focused on the methodology underlying earlier versions of platforms aimed at sharing knowledge about the connectivity of marmoset cortical areas. It included a much reduced dataset (9 animals, 17 injections), in comparison with the present one. No quantitative results of the connectivity patterns (FLNe, SLN) were available, and the *analysis* part of the portal did not exist. Furthermore, the section viewer offered no functionalities for navigating the available injections, except for just listing them. The web site associated with Majka et al. 2016 has been redesigned and reimplemented as marmosetbrain.org, where the curated datasets will continue to be deposited and integrated with the analysis interface for the foreseeable future (including the upcoming results of new experiments). Likewise, as explained above, the Lin et al. (2019) is a methods-focused paper, which describes in detail the implementation of a data processing pipeline optimised for volume reconstruction based on high-quality images obtained with the Nanozoomer instrument.

In general, it would be helpful to explicitly state (and do so earlier in the manuscript) what the present manuscript adds to the aforementioned paper, which also directs readers to marmosetbrain.org. As it stands, this is not stated until the 3rd paragraph of the introduction that features were added and additional data was made open source.

We have accepted this suggestion, by recasting the Introduction of the present version as indicated by the reviewer.

2. Although the connectivity profiles in matrix format are somewhat useful for comparing to magnetic resonance imaging techniques (e.g., rsfMRI and DTI), allowing users to directly download the tracer data in 3D volume format (e.g., a single nifti file for each injection with connectivity strengths) in a template space would be much more useful and allow for ready comparisons to topologies acquired by MRI. I would strongly urge the authors to consider this possibility. Having a downloadable structural volume (e.g., like that provided with the NIH marmoset template) to register to would also aide in this effort.

In response to this remark, we have implemented in the Resource the facility to download the results of the injections as 3D Nifti images. They can be accessed in two ways: per individual injection using the *Download results as 3D image* link, as well as in bulk via the *Results in 3D volume format (NIFTI)* link in the *Quick Links* section at the top of the page.

The results are represented as 200 μm isotropic 3D images in stereotaxic space of the Paxinos et al. (2012) template. This voxel size is dictated by the spacing between the fluorescent sections and is smaller than a typical resolution of a marmoset fMRI or diffusion imaging study (350–500 μm) which makes the 3D maps adequate, in terms of resolution, for comparing to magnetic resonance imaging.

Three variants of Niftii files for each injection are available: one may access only infragranular cells, only supragranular cells, and all labelled neurons combined. Furthermore, the package with results of all injections contains also the digitized version of Paxinos et al. (2012) template as well as its parcellation into cortical areas. To reflect these changes, we have modified the Fig. 4 as well as the relevant parts of the body of the manuscript.

Reviewers' Comments:

Reviewer #1:

Remarks to the Author:

I continue to believe this is an outstanding paper and major contribution to the field. I recommend to move forward. There is one minor wording / labeling I would suggest altering, but leave this to the discretion of the authors.

In Figure 7, a network is labeled as DMN-B. This presumably is in reference to the distinction between A and B as outlined in Braga and colleagues work in human. DMN-B in the human displays coupling to frontopolar regions and the most rostral regions of temporal association cortex, near to the temporal pole. In this sense, in the human, A and B do not differ in relation to both having components near to the rostral apex of prefrontal or temporal cortex. DMN-B labeled in figure 7 does not obviously have those characteristics as present in the human. (This does not pertain to DMN-A which as shown in Figure 7 and labeled has the multiple features expected from the human).

While I understand there may be debate about whether or not there is a candidate in the marmoset of DMN-B, and whether that candidate may deemphasize the frontal pole, it would seem a very strong statement to label it as such here uniquely, given the absence of some of the hallmark features that define it in the human. Moreover, an alternative has been published in the marmoset from the lab of Stefan Everling. In Ghahremani et al. (2018) Cerebral Ctx they illustrate in Figure 1 a series of possible network homologies between human, macaque, and marmoset. In their scheme, the present Figure 7's DMN-B may be more similar to their labeled "CON" or control network. Thus, at the very least, there is some debate and discussion to be had about which networks are truly homologous.

What I recommend is a simple fix. In Figure 7 label "DMN-B" as " DMN-B / CON" and use this as an opportunity to leave open further analysis to determine the full extent of the homologies, and that these are hypotheses (either as a line in the text, figure legend, or both). This way, the data can be presented as it is, and the varied ideas in the literature that include the Everling hierarchy are still left as open possibilities (I would also suggest citing Everling's work, since it seems relevant). This allows the present data to be displayed and fully showcases the resource, but leaves open the connections to the multiple frameworks that are evolving.

Reviewer #2:

Remarks to the Author:

Based on the replies by the authors I consider my concerns as resolved.

Reviewer #3:

Remarks to the Author:

The authors have addressed all my previous comments. I would like to thank the authors for preparing nifti files. This will be extremely valuable for the field!

Stefan Everling

Reviewer 1: Reviewer 1 identified a possible issue with the homology between resting state networks identified in the marmoset in the revised paper, and the human default mode network (DMN). He/ she proposed a workaround involving simple re-labeling of the networks in Figure 7.

We agree with the reviewer. Essentially, in our view, the problem emerges from the fact that the two studies which have attempted to identify the marmoset DMN, to date, have reached somewhat different conclusions. Our initial attempt was to label the two DMN candidates DMN-A and DMN-B according to the suggestions made by Liu et al. 2019¹. However, as correctly pointed out by the reviewer, this is unsatisfactory given the results of Braga et al., in the human brain².

To circumvent this issue, we have relabeled the DMN candidate proposed by Liu et al. CON (“cognitive control network”), following the usage proposed by the Everling group³, and the reviewer suggestion. This network does not involve the frontal pole (area 10). We also indicate in the text that this network has been proposed as a homologue of the human DMN-B, by Liu et al.¹.

In addition, to keep the nomenclature used in the paper internally consistent, we also re-labeled the network involving the frontal pole (area 10) APEX (the “apex transmodal network”)⁴. Again, the text indicates that this network has been proposed to correspond to the human DMN-A by Liu et al.¹.

These changes were incorporated to Figure 7, and are reflected in the text and figure legend.

Reviewer 2: no additional requests or comments.

Reviewer 3: no additional requests or comments.

- 1- Liu C. et al. Anatomical and functional investigation of the marmoset default mode network. *Nat. Commun.* 10, 1975 (2019).
- 2- Braga RM, Buckner RL. Parallel Interdigitated Distributed Networks within the Individual Estimated by Intrinsic Functional Connectivity. *Neuron.* 95,457-471 (2017).
- 3- Ghahremani M, Hutchison RM, Menon RS, Everling S. Frontoparietal functional connectivity in the common marmoset. *Cereb. Cortex* 27, 3890–3905 (2017).
- 4- Buckner RL, Margulies DS. Macroscale cortical organization and a default-like apex transmodal network in the marmoset monkey. *Nat. Commun.* 10, 1976 (2019).